# An end-to-end Complex-valued Neural Network approach for *k-space* interpolation in Parallel MRI

Poornima Jain                                  POORNIMA.JAIN@IIITB.AC.IN

Neelam Sinha                                   NEELAM.SINHA@IIITB.AC.IN

G. Srinivasaraghavan                           GSR@IIITB.AC.IN

*International Institute of Information Technology Bangalore*

## Abstract

Parallel MRI techniques in the *k-space*, like GRAPPA are widely used in accelerated MRI. Recently neural-network approaches have shown improved performance over linear methods like GRAPPA. But present day neural networks are largely tailored to process real data, hence the complex-valued *k-space* data is processed as two-dimensional real data in these. In this work, we study the performance of an end-to-end complex-valued architecture for interpolating missing values in the *k-space* for parallel MRI which we call the *Complex rRAKI*. We propose a novel activation function, the *PlaneReLU*, which is a generalized version of the ReLU on the complex plane. The performance of the Complex rRAKI is evaluated on two publicly-available *k-space* MRI datasets, the fastMRI multicoil brain and knee datasets. Comparison of obtained results with those on the baseline rRAKI are also presented. The proposed Complex rRAKI achieves improved performance over the baseline with respect to standard metrics SSIM and NRMSE with 50% fewer parameters.

**Keywords:** Complex-valued neural network, Complex ReLU, Parallel MRI, GRAPPA

## 1. Introduction

Parallel MRI uses multiple acquisition coils to acquire partial *k-space* data and then exploit the position information of the coils to obtain high quality reconstructions from the data. The GRAPPA (Griswold et al., 2002) method estimates missing values in the *k-space* by assuming them to be linearly-dependent on neighboring acquired values. Recently, scan-specific neural network approaches (Zhang et al., 2019; Arefeen et al., 2021) have been developed to learn a potentially non-linear relationship instead. However real-valued neural networks may not be able to exploit information in inherently complex-valued datasets. As MRI sensor data is complex-valued, few works have adapted complex-valued neural networks to MRI reconstruction (Cole et al., 2020; Chatterjee et al., 2021; Vasudeva et al., 2020). But most of these works denoise poor quality zero-filled reconstructions. When the zero-filled reconstructions have a lot of artefacts, there may be loss of important details which cannot be reconstructed back. Thus, it is important to work with *k-space* directly. Also to the best of our knowledge, none of the previous works explores a scan-specific complex-valued neural network, thus relying on huge datasets for training. In this work, we implement an end-to-end complex-valued neural network for the scan-specific Residual RAKI (Zhang et al., 2019) approach. The major contributions in this work are -

1. We implement an end-to-end complex-valued neural network trained using complex-valued optimization, called the *Complex rRAKI* for a scan-specific approach for parallel MRI called the Residual RAKI (rRAKI)

2. We propose a novel activation function, the *PlaneReLU*, which is a generalized version of the ReLU activation function on the complex plane

## 2. Methods

Let $I \in \mathbb{C}^{H \times W \times \bar{C}}$ be an input *k-space* of shape $(H, W, \bar{C})$ where $H$ and $W$ refer to the height and width and $\bar{C}$ is the number of coils, also known as number of channels. Let $AC_j$ denote the autocalibration region in channel $j$, $j = 1, 2, ..\bar{C}$. Let $y_j$ denote the vector containing target elements at locations $(k_x, k_y) \in AC_j$ and $y_{source}$ denote the vector containing the corresponding neighboring elements in the autocalibration regions across all channels. Then Complex rRAKI is trained over the autocalibration region by minimizing the cost function

$$\mathcal{L}(\gamma_j, \theta_j) = \min_{\gamma_j, \theta_j} \|y_j - G_j(y_{source}; \gamma_j) - F_j(y_{source}; \theta_j)\|_2 + \lambda \|y_j - G_j(y_{source}; \gamma_j)\|_2 \quad (1)$$

where $G_j \colon \mathbb{C}^n \to \mathbb{C}^m$ is a linear complex convolution operator parameterized by $\gamma_j$ and $F_j \colon \mathbb{C}^n \to \mathbb{C}^m$ is a complex-valued CNN parameterized by $\theta_j$, and having two blocks of complex convolution and *PlaneReLU* (Section 2.1) activation function followed by a complex convolution operation. $\mathcal{L}(\gamma_j, \theta_j) : \mathbb{C} \to \mathbb{R}$ is real-valued as it computes the $L2$ norm between the complex output and target. After learning $G_j$ and $F_j$ for each channel $j$, the interpolation for the vector of missing values $s$ in the channel $j$ is performed as

$$s(k_x, k_y, j) = G_j(\mathcal{N}(k_x, k_y)) + F_j(\mathcal{N}(k_x, k_y)) \quad (2)$$

where $\mathcal{N}(k_x, k_y)$ denotes the neighborhood for the corresponding missing point $(k_x, k_y)$ across all channels.

### 2.1. Proposed Complex-valued Activation Function : the *PlaneReLU*

We propose a version of ReLU defined over the complex plane called the *PlaneReLU*. For an input $z = x + iy \in \mathbb{C}$ i.e., $x, y \in \mathbb{R}$, the *PlaneReLU* is defined as follows

$$PlaneReLU(x + iy) = \begin{cases} x + iy, & \text{if } ax + by + c \geq 0 \\ \frac{a+b+c}{\alpha}(x + iy), & \text{otherwise} \end{cases} \quad (3)$$

where $a, b, c \in \mathbb{R}$ are learnable parameters and $\alpha$ is a hyperparameter that we set to 3. The *PlaneReLU* activation function divides the complex plane into two halves about the line $ax + by + c = 0$. It fires the input as is in one half of the plane and fires a scaled version of the input in the other half. The parameters $a, b$ and $c$ are learnt to define a suitable line according to the training dataset. The *PlaneReLU* considers both magnitude and phase information while firing without any bias towards either, unlike other ReLU-inspired complex-valued activation functions like the *zReLU* and *modReLU* (Trabelsi et al., 2018). It also does not distort the input phase, unlike the CReLU (Trabelsi et al., 2018).

## 3. Results, Discussion and Conclusions

The performance of rRAKI and Complex rRAKI architectures on the fastMRI multicoil brain and knee datasets (Zbontar et al., 2018) are presented in Table 1 and Figure 1. By

Table 1: Average PSNR, NRMSE and SSIM metrics for *k-space* data from fastMRI multicoil brain and knee datasets at an acceleration factor of 5 with cartesian undersampling

| fastmri multicoil Brain dataset | | | |
|---|---|---|---|
| Metric | PSNR | NRMSE | SSIM |
| rRAKI | $31.51 \pm 1.3$ | $0.20 \pm 0.041$ | $0.84 \pm 0.036$ |
| Complex rRAKI | $31.83 \pm 0.79$ | $0.23 \pm 0.08$ | **$0.87 \pm 0.027$** |
| fastmri multicoil Knee dataset | | | |
| rRAKI | $28.7 \pm 0.73$ | $0.45 \pm 0.09$ | $0.60 \pm 0.07$ |
| Complex rRAKI | $29 \pm 0.49$ | **$0.35 \pm 0.047$** | **$0.67 \pm 0.05$** |

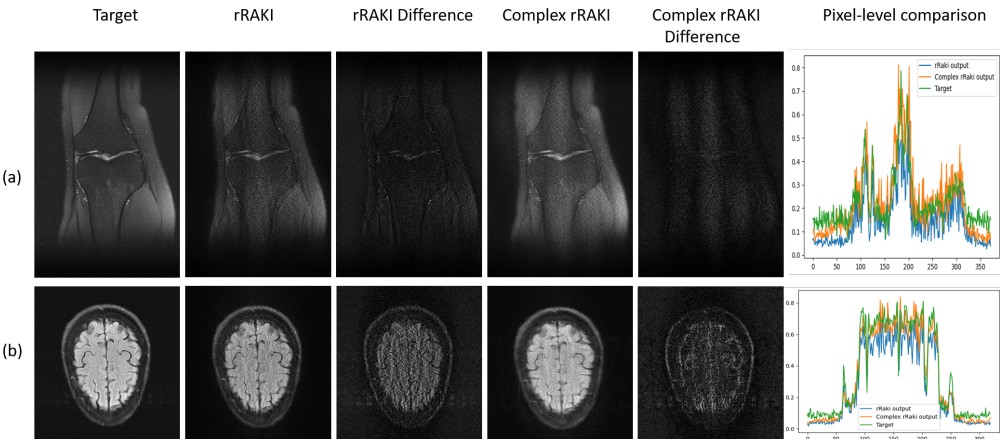

Figure 1: Sample reconstructions and difference images of rRAKI and Complex rRAKI on (a) fastMRI Knee dataset and (b) fastMRI Brain dataset. The pixel-level comparison of the $300^{th}$ index in the output is also shown.

achieving improved or comparable performance with the SOTA methodology w.r.t. SSIM, PSNR and NRMSE metrics, with 50% fewer parameters (the complex convolution layer has 50% fewer parameters than the corresponding real layer (Jain et al., 2022)), Complex rRAKI, along with its novel *PlaneReLU* activation function, shows promising potential for exploring complex-valued neural networks in the *k-space* domain as well as other such complex-valued domains. The improved performance of Complex rRAKI is attributed to the structure of its network which respects the complex-valued algebraic structure of the input, thus constraining the degrees of freedom in the neural network and assisting improved learning.

**Source code:** *https://github.com/jain-p9/Complex-rRAKI*

**Acknowledgement:** The authors gratefully acknowledge the financial support of this project through the Mphasis F1 foundation.

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
