# OpenReview forum: "An end-to-end Complex-valued Neural Network approach for k-space interpolation in Parallel MRI"
_MIDL.io/2023/Short_Paper_Track — MIDL 2023 Short paper track Poster_

### Official Review · Reviewer_3spJ · 2023-04-24
**This work proposes a complex-valued neural network, Complex rRAKI, which is specifically designed for interpolating missing values in the k-space of parallel MR imaging. The Complex rRAKI is trained using a complex-valued optimization. Experimental results show that the Complex rRAKI achieves significantly better performance than the baseline model in terms of standard metrics such as SSIM and NRMSE.**

**Rating:** 9
**Confidence:** 3

**Review:**

Overall, the idea of this proposed method is novel and interesting. Experimental results show that the Complex rRAKI achieves improved performance with 50% fewer parameters. This suggests that the proposed algorithm is a more efficient and cost-effective option for practical applications.

Strength:

-The paper is well written and organized; hence is easy for readers to follow.

-The proposed work has contributed towards understanding how to efficiently maintain both information for complex-valued optimization. Moreover, the proposed solution has important implications for improving the quality and efficiency of different medical imaging tasks via deep learning, specifically for interpolating missing values in parallel MR imaging.

-The authors' proposed an interesting version of the ReLU activation function on the complex plane, called PlaneReLU, seems a powerful addition to the well-studied field of complex-valued convolutional neural networks. By considering both magnitude and phase information, the newly designed activation function is able to effectively map learned complex-valued signals to a linear hyperplane.

Weakness:

-The introduction could benefit from a more comprehensive discussion of existing literature on complex-valued optimization.

---

### Official Review · Reviewer_yBUv · 2023-04-25
**Complex data architecture with domain-specific activation functions for MRI reconstruction**

**Rating:** 4
**Confidence:** 4

**Review:**

The authors propose a deep learning architecture for k-space data interpolation, including a non-linear activation function. While this is an important problem, the authors seem to ignore the current state of the field where pretty much all leading methods use fully complex data and appropriate activation functions tailored to the inherently complex MRI signals. Off the top of my head, below are two examples papers that explicitly describe their treatment of complex MRI data, but these are by no means the only ones. Most papers in MRI reconstruction today handle complex data and use custom activation functions. It is not surprising that the reported accuracy of reconstruction is substantially lower than the leading methods for fast MRI data the authors use to demonstrate their approach.

Kerstin Hammernik,Teresa Klatzer,Erich Kobler,Michael P Recht, Daniel K Sodickson, Thomas Pock,and Florian Knoll. Learning a variational network for reconstruction of accelerated MRI data.MagneticResonanceinMedicine,79(6):3055–3071,2018.

Nalini M Singh, Juan Eugenio Iglesias, Elfar Adalsteinsson, Adrian V Dalca, Polina Golland. Joint frequency and image space learning for MRI reconstruction and analysis. The Journal of Machine Learning for Biomedical Imaging, 2022-018, p 1-28, 2022.